# Nutrition and Flavor Evaluation of Amino Acids in Guangyuan Grey Chicken of Different Ages, Genders and Meat Cuts

**DOI:** 10.3390/ani13071235

**Published:** 2023-04-02

**Authors:** Lingqian Yin, Mingxu Xu, Qinke Huang, Donghao Zhang, Zhongzhen Lin, Yan Wang, Yiping Liu

**Affiliations:** 1Farm Animal Genetic Resources Exploration and Innovation Key Laboratory of Sichuan Province, Sichuan Agricultural University, Chengdu 611130, China; 2Guangyuan Municipal Bureau of Agriculture and Rural Affairs, Guangyuan 628000, China

**Keywords:** chicken, meat, flavor, nutrition evaluation, amino acid score

## Abstract

**Simple Summary:**

Amino acids are the substrate of protein, and also participate in the formation of flavor. Through the type and content detection of amino acid, we obtained information about meat quality characteristics of Guangyuan grey chicken in different groups of age, gender and meat cut. The content of total protein and essential amino acid in pectoral muscle of Guangyuan grey chicken was higher. The amino acid score of most essential amino acids in Guangyuan grey chicken was higher than 100, excluding Leu and Val. The content of free amino acids was higher in 120 d chicken, which was higher in roosters than in hens, and higher in thigh muscle than in pectoralis. These results provide more information for understanding the characteristics of Guangyuan grey chicken.

**Abstract:**

The composition and content of amino acids in foodstuffs have a vital impact on the nutritional value and taste. With the aim of understanding the nutrition and flavor of Guangyuan grey chicken, the composition and content of amino acids in the pectoralis and thigh muscle of chickens at the age of 90 d, 120 d and 150 d were determine using liquid chromatography–tandem mass spectrometry (LC-MS/MS) and an amino acid analyzer. A total of 17 amino acids were detected both in pectoralis and thigh muscle via the amino acid analyzer, of which the content of glutamate was the highest. Additionally, 21 deproteinized free amino acids were detected via LC-MS/MS. Among all samples, the content of glutamine in thigh muscle was the highest. The content of histidine in the pectoralis was the highest. In terms of the flavor amino acids (FAAs), the umami-taste and sweet-taste amino acids were higher in the thigh muscle of 120 d male chicken. From the perspective of protein nutrition, the essential amino acid was higher in pectoral muscle, and the composition was better. The results of the amino acid score showed that the content of leucine and valine were inadequate in Guangyuan grey chicken. Collectively, the content of amino acid in Guangyuan grey chicken was affected by age, gender and meat cut. This study confirms that meat of chicken in different ages, genders, and cuts presents different nutritional values and flavors owing to the variation of amino acids content.

## 1. Introduction

Meat is an important source of protein for humans. Of all livestock, chicken is deemed as a healthy option, with high content of protein and low content of fat, which is prevalent in different areas. The Chinese population consumes chicken in large quantities every year. Especially aged chicken, which consumers believe would perform better in nutrition and flavor. This purchase propensity is also manifested in gender. The previous studies in our team have reported that age and sex affect the meat quality of chicken [1,2]. 

The content and quality of protein are critical factors to determine the nutritional value of meat. Previous results have shown that protein quality is usually assessed by analyzing the composition and content of amino acids. The amino acid score (AAS) recommended by the WHO/FAO/UNU committee [3] is a widely used method for protein quality evaluation. The protein quality was evaluated through comparing the AAS of the tested sample against WHO/FAO/UNU standard AAS. Therefore, the content and proportion of amino acids become the crucial indicators to appraise the flavor and nutritional value of meat [4]. In animals, an imbalanced proportion of amino acid will affect the utilization of amino acids and the absorption of other nutrients [5,6], and even lead to a series of health problems [7]. Chicken is an accessible protein source with a balanced composition of essential amino acids (EAAs). It contains nine EAAs for humans, especially a high level of lysine [8,9]. The composition and content of amino acids fluctuated dynamically. There are many factors that affect the composition and concentration of amino acids in meat, such as breed, age, diet, sex, and similar [10,11,12,13]. However, whether and how these factors affect the content of free amino acids and total amino acids (TAAs) is obscure. 

The free amino acids in cells can participate in the formation of flavor substances through the Maillard reaction [14]. For instance, cysteine and glycine are important ingredients involved in the Maillard reaction to form meat flavor [15]. The flavor of meat is mainly formed by a variety of flavor precursors, which are composed of flavor nucleotides, free amino acids, soluble sugar, peptides, organic acids, salts and others [16]. Free amino acids are amongst the central flavor substances of meat, with sour, sweet, bitter, salty, monosodium L-glutamate-like (MSG-like) and other tastes. Glutamic acid (Glu) and aspartic acid (ASP) have a mixture of sour and MSG-like tastes. Serine, glycine, alanine, etc. show a sweet taste. Histidine, valine, arginine, etc. taste bitter [17]. Subsequent studies demonstrated that amino acid mixtures, or combinations of amino acids and other substances, contributed more to the umami taste of meat than individual amino acids. As revealed in an earlier study, free Glu and Y-IMP with some sour and salty ingredients were better able to imitate the flavor of bouillon [18].

Guangyuan grey chicken is a slow-growing indigenous breed, which has an average weight about 0.62 kg at 10 weeks of age. This breed is a famous indigenous breed for its grey feathers and skin. It is popular among local consumers based on the tenderness and succulence of the meat. Nonetheless, limited research has been done on this breed, resulting in indistinct data about its characteristics. Herein, to better understand the effect of the aforementioned three factors on the composition and content of amino acid in meat, we determined the composition and content of deproteinized free amino acids and total amino acids in the pectoral and thigh muscle of Guangyuan grey chickens of different ages and genders by liquid chromatography–tandem mass spectrometry (LC-MS/MS) and an amino acid analyzer, respectively. The results provide novel insights as to the nutritional value and flavor of Guangyuan grey chicken in different ages and genders, and may supplement information on the characteristics of this breed.

## 2. Materials and Methods

This trial was conducted under the guidelines approved by the Institutional Animal Care and Use Committee of Sichuan Agricultural University (approval no. -DKY2021102011). All research work was conducted in accordance with the guidelines of Sichuan Agricultural University (SAU) Laboratory Animal Welfare and Ethics.

### 2.1. Animals and Sampling

The animals for this trial were raised in Sichuan Tianguan Ecological Agriculture and Husbandry Co., Ltd. (Guangyuan, China) from October 2021 to May 2022. A total of 252 one-day-old Guangyuan grey chickens were rearing on the floor, with free access to commercial brood feed (Guangyuan Zhuangniu Agriculture and Animal Husbandry Technology Co., Ltd, Guangyuan, China) and water. At the age of 10 weeks, all the birds were divided into two groups according to gender and then transferred to a single cage (40 cm × 40 cm × 22.5 cm). Birds were given the same feed (Table 1) and kept under the same conditions, with 20–25 ℃ ambient temperature and 65% humidity. The body weight of 30 birds (half male and female) were recorded every interval Monday morning after 12 h fasting. At the age of 90 d, 120 d and 150 d, 12 birds (6 males, 6 females) were selected randomly to collect pectoralis major muscle and thigh muscle. All the samples were stored at −20 ℃ temporarily until further analysis.

### 2.2. Proximate Composition

The crude fat (CF) and crude protein (CP) of right pectoralis and thigh muscle were tested according to the National Standards for Food Safety of China (GB 5009.5-2016, GB 5009.6-2016). The moisture was determined by the oven drying method. Briefly, about 20 g of meat sample was excised after removing fascia and adipose tissue, then placed in dry petri dishes, weighed and recorded as A1. The samples were placed in an oven at 105 ℃ for 16 h, cooled to room temperature, weighed and recorded as A2. They were then put into an oven at 105 °C for 2 h again, cooled to room temperature, weighed and recorded as A3. If the |A3 − A2| ≤ 0.01 g, the water was considered to be dried. The moisture was calculated as follows:$$Moisture=\frac{A1-A3}{A1}$$

### 2.3. Sample Preparation and Amino Acid Profile

The composition and content of total amino acids in the meat were determined as portrayed by the National Standards for Food Safety–Determination of amino acids in food of China (GB 5009.124-2016). Specifically, 15 mL 6 mol/L hydrochloric acid was added to the sample after drying and crushing in a hydrolysis tube. The tube was put into the refrigerant for 3 min to freeze, then vacuumized (close to 0 Pa). The tube was filled with nitrogen and sealed, placed in a 110 ℃ incubator for hydrolysis for 22 h, cooled and transferred to a 50 mL volumetric bottle, fixed to the scale with ultrapure water, and shaken. It was dried under reduced pressure, and dissolved in sodium citrate buffer solution. After filtering, the solution was subjected to the amino acid analyzer (Hitachi L-8900, Japan) for total amino acid determination. The AAS was calculated as described in Millward’s report [19]. In the formula: M1 is the mg of amino acid in 1 g of test protein; M2 is the mg of amino acid in the reference pattern.
$$AAS=\frac{M1}{M2}\times 100\%$$

As for the deproteinized free amino acids of the pectoralis and thigh muscle, the sample preparation referred to the method of Virág D. et al [20] and Fuertig R. et al [21]. About 50 mg of fresh meat sample was weighed into a 2 mL EP tube, and 600 μL 10% methanol formate solution–H2O (1:1, V/V) was accurately added. Subsequently, 2 steel balls were added and vortex oscillation performed for 30 s. Then, it was ground at 60 Hz for 90 s, centrifuged at 12000 rpm for 5 min at 4 ℃, 20 μL of supernatant was taken and 380 μL 10% methanol–H2O (1:1, V/V) solution was added, vortex oscillation performed for 30 s, and a 100 μL diluted sample taken. The 100 μL of 100 ppb isotope label was added and vortex oscillation was performed for 30 s. The supernatant was filtered through a 0.22 μm membrane, and the filtrate was added to the detection bottle. The LC-MS/MS analysis was implemented to detect the deproteinized free amino acid concentration. The analysis was carried out on a ACQUITY UPLC system (Waters, USA) connected to an AB400 (ABS Sciex, USA) triple quadrupole mass spectrometer equipped with electro spray ionisation (ESI) probe in Sichuan Panomix Biomedical Tech Co., LTD (Chengdu, China). 

### 2.4. Statistical Analysis

Excel 2021 (Microsoft, Redmond, WA, USA) and SPSS 26 (SPSS Inc., Chicago, IL, USA) were used for data sorting and analysis. The three-way ANOVA (3 × 2 × 2 factors) was performed to evaluate the effect of age, gender and meat cut on the content of amino acid. Student’s t test was applied to analyze the difference of amino acid content between pectoralis and thigh. Post-hoc pairwise contrast was implemented to assess the difference of amino acid content among age and gender groups. The data were presented as mean ± standard error of mean (SEM).

## 3. Results

### 3.1. Body Weight and Proximate Composition 

The body weight of Guangyuan gray chickens from birth to 22 weeks of age were recorded. As shown in Figure 1, the rapid growth phase of body weight was four to six weeks of age. After six weeks of age, the difference of body weight between male and female chickens gradually increased. At the stage of sexual maturity, the weight of the rooster is about 1.7 kg (around 20 w) and the weight of the hen is about 1.5 kg (around 21 w). The proximate composition of pectoralis and thigh muscle of Guangyuan grey chickens of different ages and genders were tested (Figure 2). In summary, the age and gender had effects on moisture, CF and CP content. For chickens of all ages and genders, the moisture was higher in thigh muscle in compared with pectoralis (*p* < 0.05), while in 90 d and 120 d male chickens, it was not statistically significant. The impact of gender on moisture was mainly manifested in the 90 d. No effect of age on moisture was observed in our present study. The CF content of thigh muscle was higher than that of the pectoralis among all ages and genders (*p* < 0.05). The CF content of pectoralis of different genders was elevated with age increasing. The highest level of CF was exhibited in thigh muscle of 120 d male chickens. The CP content of all samples decreased gradually with the increase in age. At the same age, the CP content of pectoralis of male chickens was higher than that in female chickens (*p* < 0.05). For chickens of all ages and genders, the CP of the pectoralis was higher than that of the thigh muscle among different ages and genders (*p* < 0.05).

### 3.2. Amino Acids Profile 

The content and composition of total amino acids and deproteinized free amino acids in pectoralis and thigh muscle of Guangyuan grey chickens with different ages and genders were determined by amino acid analyzer and LC-MS/MS. A total of 17 amino acids were detected both in pectoralis and thigh muscle via amino acid analyzer (Appendix A), of which the content of Glu was the highest, followed by Gly and Arg. In addition, 21 free amino acids were detected via LC-MS/MS (Appendix A). Among all the samples, the content of Gln in thigh muscle was the highest, followed by Asn, His and Ala. The content of His in the pectoralis was the highest, followed by Asn, Ala and Lys. 

The variance analysis of three factors (age, gender, meat cut) was conducted for total amino acid and free amino acid. As shown in Appendix A, age had an impact on the content of nine amino acids (*p* < 0.05), gender affected the content of six amino acids (*p* < 0.05). The effect of meat cut on the content of amino acids was the most: 11 amino acids showed statistically significant differences in content caused by meat type (*p* < 0.01). The results showed that the interaction of various factors had no effect on the content of most amino acids. Regarding the effect of three factors on the content of free amino acids (Appendix A), age affects the content of 15 free amino acids (*p* < 0.01). Gender had an impact on all free amino acid content (*p* < 0.05), except GABA. Meat cut had an effect on the content of 17 free amino acids (*p* < 0.05). Part of free amino acids were affected by the interaction of different factors.

### 3.3. Amino Acid Nutrition Evaluation

Protein in food is the main source of EAAs for human body. For the nutrition assessment of Guangyuan grey chicken, the total amino acids in the pectoralis and thigh muscle were measured and the essential and non-essential amino acids (NEAAs) were compared among different groups (Table 2). Likewise, the variance analysis of three factors (age, gender, meat cut) on EAA, NEAA and TAA was conducted. As shown in Table 3, age and meat cut had an impact on the content of EAA, NEAA and TAA (*p* < 0.05). The interaction of different factors had no effect on all types of amino acids (*p* > 0.05).

The content and proportion of EAAs were analyzed to evaluate the nutritional value of protein in muscle. The pectoralis of 150 d hens exhibited the highest content of EAAs, follow by the pectoralis of 150 d roosters. There was an uptrend of EAA content with increasing age. The content of EAAs in 150 d female chicken was significantly higher than that in 90 d (*p* < 0.05). The gender had no significant effect on the content of EAAs. The content of EAAs in pectoralis was higher than that in thigh muscle. In particular, the content of EAA in pectoralis was higher than that in thigh muscle of 150 d roosters and 120 d hens (*p* < 0.05). In regard to NEAAs and TAAs, it seems to have the same variation trend as EAA. The pectoralis and thigh muscle of 150 d chicken exhibited higher content of TAAs. The content of TAAs in 150 d pectoralis of hens and roosters were higher than that in 90 d (*p* < 0.05). The ratio of EAA to TAA was not significantly different among groups.

The amino acid score represents the protein quality of food. Through comparing the content of EAAs with the reference amino acid pattern of the WHO/FAO/UNU report [3], we calculate the score of each essential amino acid to evaluate the nutritional value of protein in chicken. As shown in Table 4, the scores of amino acids were all higher than 100 except for Leu, Val and part of Lys. The score of aromatic amino acids (AAA) in thigh muscle was higher, and the score of Thr in pectoralis was higher. The scores of sulphur amino acids (SAA) were higher in both pectoral and thigh muscle. The content of Leu in pectoralis was the lowest, and the lowest score in thigh muscle was Val. Hence, the content of Leu and Val were inadequate in Guangyuan grey chicken. There was no significant difference in the scores of these two amino acids among age and gender.

### 3.4. Amino Acid Flavor Evaluation

The free amino acids in food present different flavors. We detected the content of deproteinized free amino acids in chicken from different groups through LC-MS/MS (Table 5). The contents of different flavor amino acids (FAAs) were classified as umami-taste amino acids (Glu, Asp, abbreviated to UTAAs), sweet-taste amino acids (Ala, Ser, Gly, Pro, Thr, Gln, abbreviated to STAAs) and bitter-taste amino acids (Arg, His, Met, Val, Ile, Phe, Trp, Leu, abbreviated to BTAAs). The variance analysis of three factors (age, gender, meat cut) on UTAAs, STAAs, BTAAs and FAAs was conducted. As shown in Table 6, the content of UTAAs, STAAs, BTAAs and FAAs were all affected by the three factors (*p* < 0.05). The interaction of different factors had different effects on different types of flavor amino acids. The interaction of age and gender influenced the content of all amino acid flavor types (*p* < 0.05). The interaction of age and meat cut affected the BTAAs significantly (*p* < 0.01). The interaction of gender and meat cut had an influence on STAAs (*p* < 0.05).

The highest content of UTAAs was detected in pectoralis and thigh muscle of 120 d roosters. There was a decreased trend of UTAAs content with the age increasing. The content of UTAAs in 90 d female chicken was much higher than that in 150 d (*p* < 0.05). For the STAAs, the highest content was also detected in pectoralis and thigh muscle of 120 d roosters. There were significant differences between different meat cuts, the content of STAAs in thigh muscle was higher than that of pectoralis (*p* < 0.05) at the same age and gender. There were 8 BTAAs, which were the most among the FAAs. The highest content of BTAAs was detected in thigh muscle of 120 d roosters and pectoralis of 150 d roosters, respectively. The content of BTAAs in the pectoralis was higher than that in the thigh muscle except for the 120 d rooster. Since UTAAs are the main components of flavor, the proportion of UTAAs were analyzed. The results showed that the highest proportion of UTAAs in pectoralis and thigh muscle was presented in 150 d and 90 d female chickens, respectively. In general, the content of FAAs was higher in 120 d chicken, which was higher in rooster than in hen, and higher in thigh muscle than in pectoralis. 

## 4. Discussion

For this indigenous breed, which has not been systematically selected, the information on growth trend is lacking. In this study, we recorded the body weight of the population from 0 to 22 weeks. The results showed that the growth rate of this breed was slower compared with the broilers [22,23], especially the first 4 weeks of growth. This may be due to the lower feed intake of this breed. Overall, it indicates that the growth performance of this breed needs to be further improved. Nutrition and flavor are important indexes to evaluate meat quality. Previous studies revealed that the breed, age, gender and meat cut have effects on the meat quality of chicken. In this study, we determined the proximate composition of pectoralis and thigh muscle of Guangyuan grey chicken with different ages and genders. The content of CF was considered to be associated with tenderness and flavor of meat [24,25]. We found that the thigh muscle had a higher moisture and CF than those of the pectoral muscle in all age groups, and this result was more pronounced in the case of female chickens. These results were also observed in Panpipat’s study, while there was no difference between the genders in their study [26]. Thus, the thigh muscle may have a better texture than the pectoralis of Guangyuan grey chicken. Considering the effect of gender, the highest CF was displayed in 120 d for male thigh muscle and 150 d for female thigh muscle, respectively. This result shows that the regular CF deposition may be different for chickens of different sexes. It was speculated that this phenomenon may be due mainly to sexual maturity, owning to the fact that the age at first egg of Guangyuan gray chickens is about 150 days. For pectoralis, the moisture and intramuscular fat in pectoralis showed an uptrend with the increase in age, which was also observed in Yuan’s study [14]. It seems that the older female chicken had a better taste. Protein content is one of the indexes for evaluating the nutritional value of meat. As observed in the results, the content of crude protein was higher in the pectoral than thigh muscle (*p* < 0.05). This was compatible with previous research [26,27]. It indicates that the protein nutritional value of the pectoral muscle is higher than that of the thigh muscle.

Amino acids are substrates for protein synthesis in organisms. In humans, there are nine amino acids that need to be provided by food, called EAAs. We identified eight EAAs in Guangyuan grey chicken via acid hydrolysis method, excluded Tryptophan. In this study, we identified that the content of EAA in the pectoral muscle was higher than that in the thigh muscle. This result was consistent with the results reported by Ou’s study [11]. Nevertheless, the EAA/TAA ratios of all samples were greater than 0.4, exceeding the recommended value in the FAO/WHO. Compared with other species, duck [28], pig [29], fish [30] and shrimp [31], the ratio of EAA/TAA in chicken was higher [9], as well as Guangyuan grey chicken. According to the results of AAS, the scores of amino acids were all higher than 100 except for Leu, Val and part of Lys. This result can be explained by breed and diet—it still needs to be further improved. 

Free amino acid is one of the substances that contributes to the flavor of meat via the Maillard reaction with reducing sugar [32]. In current research, the deproteinized free amino acid of the chicken was detected via LC-MS/MS to evaluate the flavor of Guangyuan grey chicken. The UTAAs were the main components of FAAs [17]. Just as the results of the free amino acid, the content of UTAA was higher in 120 d male chicken, the lowest was present to 150 d female chicken. This indicated that the content of UTAA in chicken was not linearly related to age. The content of Glu was the highest of all free amino acids, which was combined with the results of other studies [33,34]. It demonstrated that Glu is one of the central substances to form meat flavor. Sweet-taste amino acids is also an important component of flavor. The range of STAAs was different in previous studies [35,36]. In this study, we referred to the results of Kirimura et al. [17]. The differences in STAAs in different meat cuts were prominent, and the content of STAAs in thigh muscle was significantly higher than that in pectoral muscle. The highest content of STAAs was also presented in the thigh muscle of 120 d roosters. Within them, the most abundant content of STAAs was glutamine, which is in agreement with the results of previous studies [37,38]. Intriguingly, apart from the pectoralis of 150 d female chicken, the content of Gln in thigh muscle was significantly higher than that in pectoralis. In the study of Imanari et al., it was also observed that the content of Gln in vastus lateralis muscle was significantly higher than that in longissimus lumborum muscle [39]. This would suggest that the thigh muscle of Guangyuan grey chicken may be better at sweet taste. We also observed that the total amount of FAAs was higher in 120 d chicken. Considering the content of EAAs and UTAAs, 120 d is more suitable as a reference for the marketable age of Guangyuan grey chicken. Moreover, the total amount of FAAs was higher in thigh muscle than in pectoral muscle, which was in agreement with the other study [11]. 

## 5. Conclusions

In this study, we compared and analyzed the proximate composition and amino acid content of skeletal muscle of Guangyuan grey chickens at different ages, genders and meat cuts. The pectoral muscle of Guangyuan grey chicken exhibited higher total protein content and essential amino acid content, which may have higher nutritional value. The content of crude fat, UTAA and STAA in the thigh muscle was higher. The results of AAS indicated that Leu and Val were inadequate in Guangyuan grey chicken. Among the three factors, the effect of the meat cuts had a greater effect on the content of total amino acids, and gender has a greater effect on the content of free amino acid in pectoral muscle and thigh muscle. 

## Figures and Tables

**Figure 1 animals-13-01235-f001:**
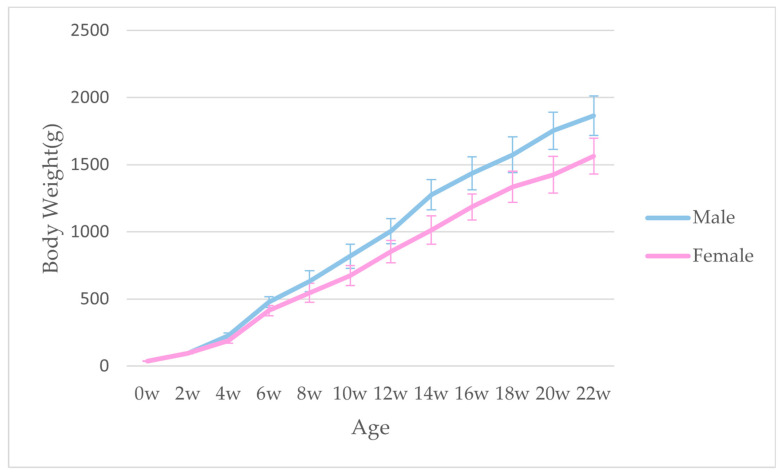
The growth trend of Guangyuan grey chicken.

**Figure 2 animals-13-01235-f002:**
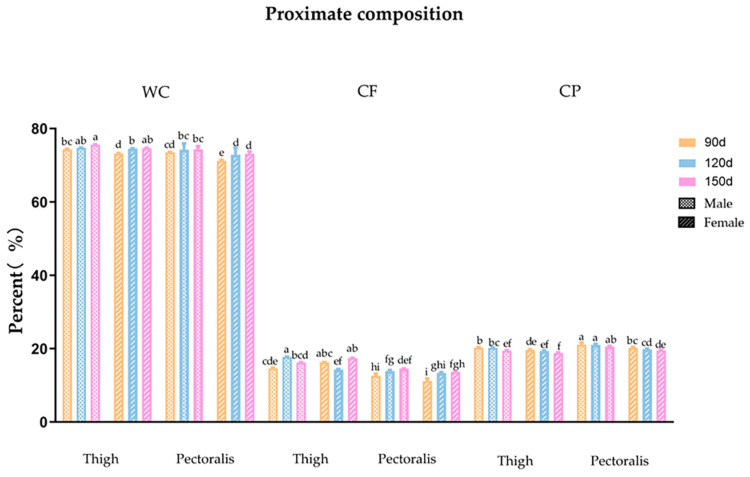
The proximate composition of Guangyuan grey chicken. WC: Water content, CF: Crude fat, CP: Crude protein. Different letters on column of the same item indicate significant difference, *p* < 0.05.

**Table 1 animals-13-01235-t001:** Composition of diet and nutrition content.

Ingredient (%)	Composition (%)
Corn	62.00
Soybean meal (sol.)	14.00
Maize gluten meal	2.00
Wheat bran	6.00
Rapeseed meal (sol.)	8.00
Soybean oil	2.00
Dicalcium phosphate	2.00
L-lysine	0.35
DL-methionine	0.15
Limestone	1.00
Premixes ^1^	2.50
Nutrition content	
ME (MJ/Kg)	11.75
CP (%)	16.45
Calcium (%)	1.09
Available P (%)	0.59
Lys (%)	1.03
Met (%)	0.43
Corn	62.00
Soybean meal (sol.)	14.00
Maize gluten meal	2.00
Wheat bran	6.00
Rapeseed meal (sol.)	8.00

^1^ Premix included the following per kg of feed: Vitamin A ≥ 390 KIU, Vitamin D3 ≥ 150 KIU, Vitamin E ≥ 1240, Vitamin B ≥ 185, Vitamin B2 ≥ 260, Vitamin B6 ≥ 160, Vitamin B12 ≥ 1.2, Vitamin K3 ≥ 100, D-biotin ≥ 12, D-pantothenic acid ≥ 470, Folic acid ≥ 57, niacin ≥ 1660, Hydrogenated choline ≥ 15,000, methionine ≥ 43,000, Fe ≥ 2000, Cu ≥ 380, Mn ≥ 3900, Zn ≥ 2800, I ≥ 25, Se ≥ 10.

**Table 2 animals-13-01235-t002:** The content and proportion of essential amino acid and non-essential amino acids (mg/g dry powder).

Items		Male			Female		
		90	120	150	90	120	150
Thigh	EAA	34.13 ± 0.89^b^	35.31 ± 1.36^ab^	36.28 ± 0.61^a^	34.31 ± 0.35^b^	35.66 ± 0.45^ab^	36.78 ± 1.90^a^
	NEAA	41.55 ± 1.44^ab^	40.88 ± 3.73^ab^	44.64 ± 2.53^a^	40.79 ± 1.27^b^	40.20 ± 0.85^b^	43.64 ± 1.53^ab^
	TAA	75.68 ± 1.59^c^	76.19 ± 4.84^bc^	80.92 ± 1.93^aa^	75.10 ± 0.93^c^	75.86 ± 0.88^c^	80.42 ± 3.32^ab^
	EAA/TAA	0.45	0.46	0.45	0.46	0.47	0.46
Pectoralis	EAA	36.38 ± 3.15^c^	38.79 ± 0.83^abc^ *	39.58 ± 0.75^ab^ **	36.87 ± 2.07^bc^	38.03 ± 0.60^abc^ **	40.13 ± 1.52^a^
	NEAA	42.08 ± 0.60^b^	43.66 ± 1.87^ab^	47.21 ± 0.66^a^	41.61 ± 1.53^b^	44.37 ± 3.68^ab^	46.86 ± 3.14^a^
	TAA	78.46 ± 3.75^b^	82.45 ± 2.66^ab^	86.79 ± 0.38^a^ *	78.48 ± 3.6^b^	82.40 ± 3.47^ab^ *	86.99 ± 1.72^a^ *
	EAA/TAA	0.46	0.47	0.46	0.47	0.46	0.46

Different letters superscript in the same row indicates significant difference, *p* < 0.05. * indicates that the same item between two cuts is significantly different at the *p* < 0.05; ** indicates that the same item between two cuts is significantly different at the *p* < 0.01. EAA, essential amino acid; NEAA, non-essential amino acids; TAA, total amino acids.

**Table 3 animals-13-01235-t003:** The variance analysis of three factors (age, gender, meat cut) of EAA, NEAA, TAA.

*p* Value	EAA	NEAA	TAA
Age	0.00	0.01	0.00
Gender	0.55	0.32	0.80
Meat cut	0.00	0.03	0.00
Age * Gender	0.64	0.73	1.00
Age * Meat cut	0.29	0.72	0.33
Gender * Meat cut	0.97	0.78	0.77
Age * Gender * Meat cut	0.74	0.95	1.00

**Table 4 animals-13-01235-t004:** Essential amino acid score of chicken.

Items		Male			Female		
		90	120	150	90	120	150
Thigh	His	169	178	191	155	168	182
	Ile	151	161	170	149	155	169
	Leu	74	73	74	72	73	74
	Lys	96	103	92	103	106	95
	Val	65	60	68	67	66	73
	Thr	154	156	160	158	163	168
	AAA	184	193	193	179	192	194
	SAA	227	234	241	223	235	248
Pectoralis	His	172	189	185	170	183	191
	Ile	173	186	194	175	183	187
	Leu	41	45	49	43	38	46
	Lys	118	115	128	127	139	144
	Val	73	78	96	80	88	99
	Thr	194	207	207	190	194	206
	AAA	142	169	154	134	157	178
	SAA	211	223	228	210	215	212

AAA, aromatic amino acids; SAA, sulphur amino acids.

**Table 5 animals-13-01235-t005:** The classification and content of flavor amino acids (ug/g fresh meat).

Items		Male			Female		
		90	120	150	90	120	150
Thigh	UTAA	441.56 ± 98.24^a^	492.49 ± 86.81^a^	330.7 ±2 5.66^abc^	487.83 ± 81.19^a^ *	425.35 ± 40.79^ab^ **	266.24 ± 29.62^c^
	STAA	1874.15 ± 262.21^ab^ *	2068.93 ± 306.66^a^ **	1877.73 ± 79.61^ab^ **	1772.09 ± 67.62^abc^ **	1640.88 ± 27.15^bc^ **	1165.41 ± 45.87^c^ **
	BTAA	853.31 ± 99.87^ab^	1133.34 ± 223.39^a^	754.39 ± 15.25^ab^	643.98 ± 58.19^bc^	736.73 ± 48.7^abc^	400.1 ± 3.94^c^
	FAA	3169.02 ± 459.22^ab^ *	3694.75 ± 603.01^a^ *	2962.81 ± 93.16^ab^ **	2903.9 ± 204.55^abc^ *	2802.97 ± 107.09^bc^ **	1831.74 ± 42.6^c^ **
	UTAA/FAA	0.1393	0.1333	0.1116	0.168	0.1518	0.1453
Pectoralis	UTAA	332.82 ± 45.81^ab^	409.99 ± 96.09^a^	311.33 ± 29.61^ab^	315.62 ± 41.7^ab^	275.76 ± 7.95^bc^	241.78 ± 27.13^c^
	STAA	896.26 ± 28.03^ab^	1123.19 ± 65.66^a^	890.73 ± 113.9^b^	921.95 ± 67.58^ab^	760.59 ± 103.87^bc^	638.15 ± 49.44^c^
	BTAA	1044.88 ± 93.05^a^	1066.29 ± 97.5^a^	1096.05 ± 122.21^a^ *	895.55 ± 56.61^ab^ **	807.3 ± 15.13^b^	626.95 ± 18.46^c^ **
	FAA	2273.96 ± 136.84^b^	2599.47 ± 253.42^a^	2298.11 ± 33.74^b^	2133.12 ± 18.64^b^	1843.65 ± 104.59^c^	1506.88 ± 40.23^d^
	UTAA/FAA	0.1464	0.1577	0.1355	0.148	0.1496	0.1605

Different letters superscript in the same row of the same classification indicates significant difference, *p* < 0.05. * indicates that the same item between two cuts is significantly different at the *p* < 0.05; ** indicates that the same item between two cuts is significantly different at the *p* < 0.01. UTAA, umami-taste amino acids; STAA, sweet-taste amino acids; BTAA, bitter-taste amino acids; FAA, flavor amino acids.

**Table 6 animals-13-01235-t006:** The variance analysis of three factors (age, gender, meat cut) of UTAA, STAA, BTAA and FAA.

*p* Value	UTAA	STAA	BTAA	FAA
Age	0.00	0.00	0.00	0.00
Gender	0.02	0.00	0.00	0.00
Meat cut	0.00	0.00	0.00	0.00
Age * Gender	0.07	0.00	0.02	0.00
Age * Meat cut	0.05	0.27	0.00	0.04
Gender * Meat cut	0.26	0.02	0.66	0.24
Age * Gender * Meat cut	0.77	0.17	0.25	0.84

## Data Availability

The data relevant to the study are available from the corresponding author upon reasonable request.

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
