# Peer review of "Nutrition and Flavor Evaluation of Amino Acids in Guangyuan Grey Chicken of Different Ages, Genders and Meat Cuts"

_animals, 2023, doi:10.3390/ani13071235_

Round 1

Reviewer 1 Report

The authors compared and analyzed the proximate composition and amino acid content of skeletal muscle of Guangyuan Grey chickens at different ages, genders and meat cuts, aiming to understand the nutrition and flavor of Guangyuan Grey chicken. The results are helpful for the nutrition and flavor evaluation of Guangyuan Grey chicken. However, I think that the present version of this manuscript is not suitable for publication in Animals. Some modifications must be made to improve the quality of the present manuscript.

Major concerns

1. In the Statistical Analysis, the authors said they performed normality and homogeneity of variance test. However, I can not find the test results in the manuscript.

2. My major concern, most Tables in the paper are hard to read. The authors combined all the three factors in a single table, markered difference using different letters, making the Tables to complicated and the result difficult to understand. For example, in Table 2, I try to analyze the effect of age on the content of crude fat. However, I find that each age stage contains four rows of results, divided into two genders and four parts. So how to compare the results? I recommend that the author must imrove the presentation of the results to make it easy to read and understand.

3. The authors indicated that they used three-way ANOVA to analyze the results. For proximate composition analysis, is the three-way ANOVA used? I can not find the significance test of gender, age and part, and the test of their interaction effects.

4. I recommen that some of the results in Tables could be changed to be present using graphs.

5. In the abstract, the author concluded that Guangyuan Grey chicken performs well in nutrition and flavor of amino acid, especially the 120d chicken. I do not think that the result can support this conclusion. At least, The authors should compared different breeds of chicken.

6. Why the authors detected different animo acids using animo acid analyzer and LC-MS/MS with the same samples?

Minor concerns

1. Guangyuan Grey chicken or Guangyuan grey chicken?

2. Line 151-153, for chickens of all ages and genders, the moisture were higher in thigh muscle in compared with pectoralis, except for 90d and 120d male chicken (p<0.05). Please confirm the results again.

3. Leg muscle or thigh muscle?

4. In the Animals and Sampling section, the authors collected pectoralis major muscle or minor muscle?

5. Line 10, delete "of".

Author Response

Responses for revision

Dear Editors and Reviewers:

Thank you for your letter and the reviewers’ comments concerning our manuscript entitled “Nutrition and flavor evaluation of amino acids in chicken of different ages, genders and meat cuts” (ID: animals-2282691). Those comments are very helpful for revising and improving our paper. We have studied comments carefully and have made correction which we hope meet with approval. Revised portion are marked in red in the paper. The responds to the reviewer’s comments are as flowing:

Responds to the reviewer’s comments:

Reviewer: 1

  1. C: In the Statistical Analysis, the authors said they performed normality and homogeneity of variance test. However, I can not find the test results in the manuscript.

R: Thank you for reminding. The normality and homogeneity of variance test of all data were conducted before proceeding to further analysis. Compared to the subsequent analysis, these results are not the main content. Therefore, it does not show in the text. To avoid misleading, we deleted this statement.

  1. C: My major concern, most Tables in the paper are hard to read. The authors combined all the three factors in a single table, markered difference using different letters, making the Tables to complicated and the result difficult to understand. For example, in Table 2, I try to analyze the effect of age on the content of crude fat. However, I find that each age stage contains four rows of results, divided into two genders and four parts. So how to compare the results? I recommend that the author must imrove the presentation of the results to make it easy to read and understand.

R: Thank you for reminding. We were also aware of this problem when preparing the manuscript. After discussion, we divided the data of gender and age for Post Hoc Multiple Comparisons, and conducted a Student's t test for the data of meat parts. We hope that the results presented in this way can be clearer.

  1. C: The authors indicated that they used three-way ANOVA to analyze the results. For proximate composition analysis, is the three-way ANOVA used? I can not find the significance test of gender, age and part, and the test of their interaction effects.

R1: Thank you for reminding. The three-factor analysis of variance was only used to analyze the data of amino acids. As for the data of proximate composition, we only conducted the Post Hoc Multiple Comparisons among different groups. The method description here is inappropriate and we have corrected it.

  1. C: I recommend that some of the results in Tables could be changed to be present using graphs.

R: Thank you for your suggestion. We have changed the results of Table 2 to graph (Figure 2).

  1. C:In the abstract, the author concluded that Guangyuan Grey chicken performs well in nutrition and flavor of amino acid, especially the 120d chicken. I do not think that the result can support this conclusion. At least, The authors should compared different breeds of chicken.

R: Thank you for reminding. We have deleted the statement in conclusion and made some modification. The comparison of different breeds of chicken will be study in the future.

  1. C: Why the authors detected different animo acids using animo acid analyzer and LC-MS/MS with the same samples?

R: Compared to proteolytic amino acids, the content of free amino acids is much lower. In order to ensure the accuracy of the results, we chose the amino acid analyzer and LC-MS/MS to detect the content and type of amino acids according to the detection sensitivity of machines.

  1. C: Guangyuan Grey chicken or Guangyuan grey chicken?

R: Thank you for reminding. We have unified the name of Guangyuan grey chicken in the full text.

  1. C: Line 151-153, for chickens of all ages and genders, the moisture were higher in thigh muscle in compared with pectoralis, except for 90d and 120d male chicken (p<0.05). Please confirm the results again.

R: Thank you for reminding. The statement here was easily confusing. We changed this statement into “the moisture were higher in thigh muscle in compared with pectoralis (p<0.05), while in 90d and 120d male chicken was not statistically significant.”

  1. C: Leg muscle or thigh muscle?

R: Thank you for reminding. We have unified the statement of thigh muscle in the full text.

10.C: In the Animals and Sampling section, the authors collected pectoralis major muscle or minor muscle?

R: Thank you for reminding. The pectoralis was sampling from pectoralis major muscle. We made it more clear in the Animals and Sampling section.

  1. C: Line 10, delete "of".

R: Thank you for reminding. We have deleted the “of” in the Simple Summary section.

We appreciate for Reviewer’ warm work earnestly, and hope that the correction will meet with approval.

Once again, thank you very much for your comments and suggestions.

Sincerely,

Lingqian Yin and co-authors.

Reviewer 2 Report

Dear authors, although this study provides new data about the amino acid profile of Guangyuan grey chicken meat, this breed is a local breed with limited importance for worldwide broiler production. Moreover, the authors did not provide performance data such as body weight (BW), body weight gain (BWG) feed intake (FI), and feed conversion ratio (FCR) to better evaluate the importance of the results. In addition, the authors did not provide the bioethical committee from which they took permission to conduct the present experiment. Thus, the discussion section is very short (only three paragraphs). Conclusively this work needs to be rejected and not be published in this journal because of its limited importance for the world poultry scientific society and due to many serious omissions. The specific parts of the manuscript in which authors can find the reasons of the decision are presented line by line in the following paragraphs.

Simple summary:

L. 15-16. In the title, the topic of the research is more general and has a greater level of importance for chicken foodstuffs. However, in simple summary authors highlight the results specific to Guangyuan grey chickens. The authors must delete this sentence in this section of the manuscript.

Abstract:

L. 29-31. Authors must write the conclusion in more general form for all chickens and not focus only on Guangyuan grey chickens.

Introduction:

L. 37-38. What is the percentage of Guangyuan chickens from the 73.79 billion chickens according to the latest FAO statistics? Authors must provide the percentage to show the importance of their work.

L. 70-75. The authors wrote that Guangyuan Grey chicken is a slow-growing indigenous breed, and it is only popular among local consumers and that due to these reasons, there is a limited number of studies done on this breed. Why authors selected this breed although it has limited importance for world poultry production.

Materials and Methods:

L. 88. According to which bioethical committee birds were euthanized. Authors must provide the protocol number of the experiment and the specific bioethical committee which gave permission to authors to conduct the present experiment.

L. 96-97. The ages of 90d, 120d, and 150d are very costly for broiler production. The breed that you have selected can not fit in the producer's and consumers' requirements worldwide but only at the local level.

Results:

----------------------------------------------------------

Discussion:

L. 260-317. The discussion section is too short (only 3 paragraphs). The authors did not provide any data about animal zootechnical parameters such as BWG, BW, FI, and FCR. Moreover, authors only focus on a specific breed which has very limited importance in worldwide broiler production.

Conclusions:

----------------------------------------------------------

Author Response

Responses for revision

Dear Editors and Reviewers:

Thank you for your letter and the reviewers’ comments concerning our manuscript entitled “Nutrition and flavor evaluation of amino acids in chicken of different ages, genders and meat cuts” (ID: animals-2282691). Those comments are very helpful for revising and improving our paper. We have studied comments carefully and have made correction which we hope meet with approval. Revised portion are marked in red in the paper. The responds to the reviewer’s comments are as flowing:

Responds to the reviewer’s comments:

Review 2

  1. C: L. 15-16. In the title, the topic of the research is more general and has a greater level of importance for chicken foodstuffs. However, in simple summary authors highlight the results specific to Guangyuan grey chickens. The authors must delete this sentence in this section of the manuscript.

R:Thank you for reminding. After discussion, we recognized that the results of this trial were based on the Guangyuan Grey Chicken. The results had a guiding significance to the other study. Hence, we corrected this issue by modifying the title into “Guangyuan grey chicken”.

  1. C: L. 29-31. Authors must write the conclusion in more general form for all chickens and not focus only on Guangyuan grey chickens.

R:Thank you for your suggestion. We referred to similar articles and made some adjustments to our conclusions.

  1. C: L. 37-38. What is the percentage of Guangyuan chickens from the 73.79 billion chickens according to the latest FAO statistics? Authors must provide the percentage to show the importance of their work.

R:Thank you for reminding. There is no relevant official statistics about the Guangyuan grey chickens. We have deleted this statement in the manuscript.

  1. C: L. 70-75. The authors wrote that Guangyuan Grey chicken is a slow-growing indigenous breed, and it is only popular among local consumers and that due to these reasons, there is a limited number of studies done on this breed. Why authors selected this breed although it has limited importance for world poultry production.

R:Thank you for reminding. As an indigenous breed, there is little information about Guangyuan grey chicken, which also indicates that underlying good qualities of this breed are yet to be discovered. In addition, Guanyuan grey chicken has the characteristics of grey coat, which is rare among chicken breeds. Therefore, we chose this breed to study and explored its economic value, which is also conducive to the subsequent promotion of this breed.

  1. C: L. 88. According to which bioethical committee birds were euthanized. Authors must provide the protocol number of the experiment and the specific bioethical committee which gave permission to authors to conduct the present experiment.

R:Thank you for reminding. We modified the Ethics statement in Materials and Methods section.

  1. C: L. 96-97. The ages of 90d, 120d, and 150d are very costly for broiler production. The breed that you have selected can not fit in the producer's and consumers' requirements worldwide but only at the local level.

R:We chose these three days of age mainly due to they are the days of age for chicken to go on sale.

  1. C: L. 260-317. The discussion section is too short (only 3 paragraphs). The authors did not provide any data about animal zootechnical parameters such as BWG, BW, FI, and FCR. Moreover, authors only focus on a specific breed which has very limited importance in worldwide broiler production.

R:Thank you for reminding. The discussion section was arranged according to the three aspect, proximate composition, nutrition and flavor of amino acid, which was consistent with the logic of the full text. The data of grow trend were added in the manuscript in Figure 1.

We appreciate for Reviewer’ warm work earnestly, and hope that the correction will meet with approval.

Once again, thank you very much for your comments and suggestions.

Sincerely,

Lingqian Yin and co-authors.

Reviewer 3 Report

The study evaluated the nutritional value and flavor of amino acids in chickens of different ages, sexes and meat cuts. The conclusions of the conducted research are clear and result from the obtained research results. The material used for the research is sufficient, the research methods have been selected appropriately. Discussing the results against the background of other authors is very detailed. The publications cited by the authors of the article are well selected. For the most part, the authors refer to the latest knowledge published in renowned scientific journals. I could not find any mistakes in the scientific aspect of the manuscript.

However, the authors did not avoid a few mistakes, which I will list below:

- A few punctuation problems are present in the manuscript. I suggest the Authors to double-check the text.

Author Response

Responses for revision

Dear Editors and Reviewers:

Thank you for your letter and the reviewers’ comments concerning our manuscript entitled “Nutrition and flavor evaluation of amino acids in chicken of different ages, genders and meat cuts” (ID: animals-2282691). Those comments are very helpful for revising and improving our paper. We have studied comments carefully and have made correction which we hope meet with approval. Revised portion are marked in red in the paper. The responds to the reviewer’s comments are as flowing:

Responds to the reviewer’s comments:

Review 3

  1. C: A few punctuation problems are present in the manuscript. I suggest the Authors to double-check the text.

R:Thank you for reminding. We have checked the text again and made some revisions.

We appreciate for Reviewer’ warm work earnestly, and hope that the correction will meet with approval.

Once again, thank you very much for your comments and suggestions.

Sincerely,

Lingqian Yin and co-authors.

Round 2

Reviewer 1 Report

I think that the authors have addressed most of my concerns. However, I found that the authors added Figure to the manuscript, which showed the growth trend of Guangyuan grey chicken. As is known, the growth curve of chicken generally follows a "S" type. The gorwth curve of Guangyuan grey chicken failed to follow the "S" type. Please explain this issue. 

Maybe the market age of Guangyuan grey chicken is 120 d?

Author Response

Dear Reviewer:

Thank you for your comments concerning our manuscript entitled “Nutrition and flavor evaluation of amino acids in chicken of different ages, genders and meat cuts” (ID: animals-2282691). Those comments which you mentioned in Round 1 were very helpful for revising and improving our paper. About the two questions you mentioned in Round 2, the responds are as follows:

1.C: I think that the authors have addressed most of my concerns. However, I found that the authors added Figure to the manuscript, which showed the growth trend of Guangyuan grey chicken. As is known, the growth curve of chicken generally follows a "S" type. The gorwth curve of Guangyuan grey chicken failed to follow the "S" type. Please explain this issue.

R: Thank you for reminding. About this phenomenon, we deduce it may be due to the breed and the feedstuff. The growth curve of chickens generally presents an "S" shape, which is mainly observed in commercial broiler breeds. To be honest, for this indigenous breed, which has not been systematically selected, we have not figured out its exact growth trend yet on account of the feeding standard are still in the exploratory stage. The only know is the slow growth rate and the brooding period is long for 10 weeks.

As for the feed, the quality is out of control and the nutritional requirements of this breed are still unknown. The birds at the brooding stage were supplied with commercial brood feed purchased from local feed company. Different batches of feed ingredients may not come from the same place. Maybe the energy and protein content of the diet during the brooding and growing period is insufficient for this breed. In addition, the low feed intake of this breed, combined with the lack of nutrition in the feed, may result in the failure to grow rapidly during the assumed rapid growth phase. In general, these reasons may affect the growth curve.

2.C: Maybe the market age of Guangyuan grey chicken is 120 d?

R: Thank you for your suggestion. Actually, the consumers prefer chickens more than 1.5kg, which in this breed is about 21w for female chicken and 17w for male chicken. However, considering the feeding cost, growth trend and meat quality, it may suitable to go on sale at the age of 120 days for female chicken and 140d for male chicken.

We appreciate for you warm work earnestly.

Best regards,

Yiping Liu and co-authors.

Reviewer 2 Report

The authors answered all my scientific concerns analytically. For this reason, I believe that this research is ready to be published in the present journal.

Author Response

Dear reviewer,

Thank you for carefully reviewing our work and providing such good suggestions. The manuscript has been improved a lot with your help.

Thank you again for your time and input in facilitating the publication of this work.

Beat regards,

Yiping Liu